

# Evaluation of hygroscopic cloud seeding in liquid-water clouds: a feasibility study

Fei Wang[1,2], Zhanqing Li[2,3], Qi Jiang[4], Gaili Wang[5], Shuo Jia[6], Jing Duan[1], Yuquan Zhou[1]

[1] Key Laboratory for Cloud Physics, Chinese Academy of Meteorological Sciences, Beijing, 100081, China
[2] State Key Laboratory of Earth Surface Processes and Resource Ecology and College of Global Change and Earth System Science, Beijing Normal University, Beijing, 100875, China
[3] Department of Atmospheric and Oceanic Science, University of Maryland, College Park, MD 20742, USA
[4] National Meteorological Center, Beijing, 100081, China
[5] State Key Laboratory of Severe Weather, Chinese Academy of Meteorological Science, Beijing 100081
[6] China Huayun Group, Beijing, 100081, China

*Correspondence to*: Zhanqing Li (zli@atmos.umd.edu)

**Abstract.** An airborne cloud seeding experiment was conducted over the eastern coast of Zhejiang, China, on 4 September 2016 during a major international event held in Hangzhou. In an attempt to reduce the likelihood of rainfall onset, a major airborne experiment for weather modification took place by seeding hygroscopic agents to warm clouds to reduce cloud droplet size. The effectiveness of seeding is examined, mainly for stratiform clouds with patchy small convective cells. A radar-domain-index algorithm (RDI) was proposed to analyze the seeding effect. The threshold strategy and the tracking radar echo by correlation (TREC) technique was applied in the domain selection. Factors analyzed include echo reflectivity parameters such as the mean and maximum echo intensity, the anomaly percentage of the grid number of effective echoes, the fractional contribution to the total reflectivities, and the vertically integrated liquid water content (*VIL*) during and after the seeding process. About 12 minutes after seeding ended, the composite reflectivity of seeded clouds decreased to a minimum (< 10 dBz) and the *VIL* of seeded clouds was ~0.2 kg m$^{-3}$. The echo top height dropped to ~3.5 km, and the surface echoes were also weakened. By contrast, there was no significant variation in these echo parameters for the surrounding non-seeded clouds. The seeded cell appeared to have the shortest life cycle, as revealed by applying the cloud-cluster tracking method. The airborne CDP measured cloud number concentration, effective diameter and liquid water content gradually increased since the cloud seeding start. This probably because the hygroscopic growth by agent particles and collision-coalescence by small cloud droplets. However, these parameters sampled at ~40 min after seeding decreased significantly, which probably due to the excessive seeding agents generated a competition for cloud water and thus suppressing cloud development and precipitation. Overall, the physical phenomenon was captured in this study, but a more quantitative in-depth analysis of the underlying principle is needed.

## 1. Introduction

Weather modification, mainly by cloud seeding, is a common technique of changing the amount or intensity of precipitation.



Cloud seeding activities include dispersing agents to a cloud by ground-based (Dessens, 1998), rockets (Warburton et al., 1982; Radinović and Ćurić, 2007), and aircraft (Jung et al., 2015; French et al., 2018). The seeding agents can serve as cloud condensation nuclei (CCN) to advance the collision-coalescence process in warm clouds (Jensen and Lee, 2008; Jung et al., 2015), or serve as ice nuclei (IN) to convert liquid water into ice crystals and strengthen vapor deposition, riming, and

aggregation processes in super-cooled clouds. The theories behind hygroscopic and glaciogenic cloud seeding have been well documented (Schaefer, 1946; Vonnegut, 1947; Bowen, 1952), but the actual effect in practice remains highly uncertain and even controversial (Council, 2003).

Many laboratory, modeling, and field experimental studies on cloud seeding have been conducted for more than a half century, assessing the effectiveness of cloud seeding is very challenging due to notorious difficulties in gaining convincing scientific

evidences. The randomized evaluation of cloud seeding based on multiple samples has been performed with relatively high support and confidence (Gagin and Neumann, 1981; Silverman, 2001). However, conducting a long-term, well-designed, and randomized cloud seeding experiment is fraught with difficulties and uncertainties (Guo et al., 2015). Relative to modeling and statistical evaluations, much fewer have been done to acquire direct observational evidences in field experiment of the effectiveness of cloud seeding (Kerr, 1982; Mather et al., 1997; Silverman, 2003). Encouraged by some recent successes

(Tessendorf et al., 2012, 2018), we have attempted to investigate the effectiveness of cloud seeding by exploring different evaluation methods.

Presented here is a study of assessing the cloud seeding effect by injecting hygroscopic particles into a convective cell in a warm stratocumulus cloud for the prevention of rainfall. Hygroscopic seeding to promote the drop collision-coalescence process in liquid-water clouds has been investigated for some time (Bowen, 1952). Rosenfeld et al. (2010) concluded that

hygroscopic seeding was generally guided by three conceptual models: seeding with large CCN that serve as embryos for raindrops, acceleration of the coalescence process via the competition effect, and widening of cloud drop size distribution though the tail effect. However, hygroscopic materials of different properties, different concentration and size distribution may have positive or negative responses to cloud seeding. Previous studies (Bruintjes, 2003; Belyaeva et al., 2013) have shown that introducing a certain amount of CCN into clouds could broaden the cloud droplet spectrum at the initial stage of condensation,

intensify coagulation during the formation of precipitation, and enhance the lifetime of convective clouds by changing their vertical structure. For example, flares generate giant hygroscopic particles which could shift the cloud drop size distribution toward large sizes, thereby promoting the coalescence process and enhancing precipitation (Tzivion et al., 1994; Cooper et al., 1997). A modeling study (Segal et al., 2004) showed that hygroscopic particles with diameters ($D$) of 3–6 μm are optimal for enhancing precipitation in liquid-water clouds. Conversely, high concentrations of small hygroscopic particles may suppress

precipitation (Rosenfeld et al., 2008). The increasing CCN from anthropogenic pollution causes higher cloud drop concentration and narrower droplet spectrum, leading to suppressed drizzle formation and prolonged stratiform clouds (Bruintjes, 2003). They also produce brighter clouds that are less efficient in precipitation (Albrecht, 1989). Some modeling studies on hygroscopic seeding have suggested similar effects, such as Yin et al. (2000) who reported that particles with $D$ less than 2 μm had a negative effect on rain development in convective clouds.



Radar observation has been used to probe any changes in cloud properties after seeding (Hobbs et al, 1981; French et al., 2018). To this end, radar-based methodologies have been developed such as the Thunderstorm Identification, Tracking, Analysis and Nowcasting (TITAN, Dixon and Wiener, 1993) and Tracking Radar Echo by Correlation (TREC, Rinehart and Garvey, 1978) were used in cloud seeding experiments (Woodley and Rosenfeld, 2004; Rosenfeld, 1987). The traditional $Z$-$R$ relationship for estimating rainfall has also been widely used in randomized cloud seeding experiments (Cunning Jr, 1976; Dennis et al., 1975). China has the world's largest operational program of weather modification (Guo et al., 2015). The airborne and ground-based instrumentation has been greatly enhanced, which helped reduce the observation uncertainties. Several field experiments were conducted in recent years for more scientific identification and quantification of the cloud seeding effects (Lu and Guo, 2012; Zhu et al., 2015), but their effectiveness remains highly uncertain with some critical issues not being resolved yet.

The goals of this study are to evaluate any consequence of aircraft hygroscopic seeding and to develop a feasible method for analyzing the cloud seeding effect for stratocumulus clouds by:

      (a) analyzing the variability of radar parameters in nearby regions with and without seeding,

      (b) tracking and comparing the lifetime between seeded and unseeded echoes;

      (c) examining the variation of surface precipitation;

      (d) analyzing the cloud microphysics before and after cloud seeding.

## 2.    Experimental and data description

Unlike the usual practice of cloud seeding that chiefly aims at triggering and enhancing rainfall, our seeding was intended for suppressing rainfall in support of the opening ceremony of the G20 Hangzhou Summit. A series of field experiments were conducted off the eastern coast of Zhejiang during August and September 2016. A ground-based Doppler S-band radar deployed at Zhoushan (122.11°E, 30.07°N, ~438 m above sea level) provided useful information for identification of seeding echoes. The volume scan pattern (VCPs) was the standard mode of precipitation observation in six-minute intervals and a minimum elevation angle of about 0.5°. A twin turboprop (Modern Ark 60, MA-60) research aircraft was responsible for cloud seeding, that was equipped with in situ probes to measure aerosol, cloud, and rainfall particles that are integrated in a system developed by the Droplet Measurement Technologies Inc. (DMT), provided cloud microphysical observations. The aircraft-mounted cloud physics probe relevant to this study is the Cloud Droplet Probe (CDP) which resolves cloud drops with D ranging from 2 to 50 μm. Details about the uncertainties of the CDP and the CDP-estimated liquid water content (LWP) in liquid-water clouds have been described in other studies (Lance, 2010, 2012; Faber et al., 2018). Air temperature was also measured to ascertain if a cloud is in liquid phase. The typical speed of the aircraft is 60–70 m s$^{-1}$ during the cloud seeding and cloud microphysics sampling operations. The frequency of data acquisition is 1-Hz in this study.

The hygroscopic cloud seeding agent used in the experiment was the ZY-1NY flare developed by Shannxi Zhongtian Rocket Technology. The combustion product of ZY-1NY flares primarily consists of potassium chloride (KCl) and calcium chloride (CaCl$_2$) which leads to the formation of accumulation- and coarse-mode hygroscopic aerosols (D > 0.5 μm) in the shape of



salt aggregates. The agent was set to disperse the hygroscopic particles at a rate of ~4.4×10$^{13}$ s$^{-1}$ under laboratory conditions. For hygroscopic particles larger than D > 0.5 μm, D > 0.9 μm, and D > 2 μm, we estimate that ~4.0×10$^{11}$, ~6.0×10$^{10}$, and ~2.0×10$^{10}$ hygroscopic particles were emitted per second, respectively. Table 1 gives details about the technical parameters of the ZY-1NY cloud seeding flare.

Marine stratocumulus clouds were observed off the coast of eastern Zhejiang on 4 September 2016. Figure 1a shows the cloud image of Himawari-8 at visible channel (0.47μm) at 0300 UTC. There was a wide range stratiform cloud over eastern China, and a severe tropical storm (Typhoon Namtheun, NO. 1612) located near Yatsushiro Sea. By the co-work of these two systems, a weak easterly wave was developing off the east coast of Zhenjiang. Inhomogeneity internal structure within a low-level cloud deck was captured from satellite image. The cloud showed apparently double-layer structure over experimental region. The
wide range continental cloud of 8~12 km height was dominated by west wind, however, the easterly wave cloud, which mainly blow ~4 km height, was dominated by east wind (Figure 1b and S1). From cloud optical depth (COD) of satellite image in Figure 1c, the large values of COD mainly contributed by the low-level cloud.

Real-time Himawari-8 satellite and ground-based radar images were used to identify cloud decks for cloud seeding purposes. Radiosondes were also launched from surrounding cities, i.e., Shanghai (121.44°E, 31.40°N; ~150 km north of the experiment area), Hangzhou (120.16°E, 30.25°N; ~150 km west of the experiment area), and Taizhou (121.41°E, 28.62°N; ~150 km south
of the experiment area) at 0000 and 0600 UTC. Routine meteorological measurements, especially rain gauge data of hourly precipitation, made by the China Meteorological Administration (CMA) are also used. Together, they provided information on the state of the atmosphere and cloud properties that are critical to the seeding experiment and the design of the sampling flight pattern in and out of clouds. The MA-60 aircraft seeded the clouds along a circular trajectory ~5 km in diameter centered at
~121.8°E, 29.8°N. Eight ZY-1NY flares were burned in the middle part of the cloud (1900 -2200 m). Since the mean wind direction was northeasterly at the seeding altitude, the seeded cloud exposed to burned flares moved toward the southwest. The aircraft sampled the seeded cloud about 15 minutes after the seeding was completed so that the seeding effect could be studied. Figure 2 shows the flight track of the aircraft and the sampling position of the seeded cloud.

## 3.    Method and Results

The ground-based S-band radar and airborne CDP data were analyzed for evaluating the efficacy of cloud seeding. Since there was a sufficient amount of water vapor in the lower atmosphere of the experimental region on 4 September 2016, the cloud seeding region was mainly covered by stratiform clouds with patchy small convective cells. The RDI method based on radar grid data is proposed to analyze the cloud seeding effect.

### 3.1 Radar-domain-index (RDI) algorithm

The radar grid data used by the radar-domain-index (RDI) method are from the Doppler Weather Radar 3D Digital Mosaic



System (RDMS), details about the RDMS have been described by Wang et al. (2009). Quality control was performed on the reflectivity data to remove electronic interference, ground clutter, and anomalous propagations. The 3D Cartesian-gridded reflectivity data were then interpolated from the spherical coordinate system (Wang et al., 2012, 2013). The interpolation method is a nearest-neighbor scheme on range-azimuth planes combined with a linear interpolation in the vertical direction.

This method has proved to be a sound scheme that retains high-resolution structures comparable to the raw data (Xiao et al., 2008). Based on high spatial-temporal resolution 3D mosaic reflectivity data, several radar features were obtained from the RDMS: the radar constant altitude plan position indicator (*CAPPI*), composite reflectivity (*CR*), and vertically integrated liquid (*VIL*). The vertical resolution of *CAPPI* was 500 m, and the horizontal resolution of the three products was 0.01°×0.01° (~1 km×1 km). The temporal resolution was six minutes.

The RDI method estimates the cloud seeding efficacy by analyzing radar echo parameters before and after seeding over same or different areas, which pending on the motion of the seeded clouds. For example, assuming that the strong echo in the black rectangle is a seeding cloud, three domains can be defined: Domain A, defined as the effective detection scope of the Doppler radar (Figure 3a); Domain B, the possible influence area of the seeded cloud based on the diffusion of agents by wind and the motion of the seeding plume (Figure 3b); and Domain C, a movable region which represents the seeding echoes at a particular

time (Figure 3c). In Domain C, a threshold value of reflectivity and the TREC technique (Rinehart and Garvey, 1978; Tuttle and Gall, 1999) are performed to identify and track the motion of the seeded echo, and the echo parameters in this domain are also calculated. Details about the TREC method are provided in the supplementary document.

Typical parameters related to the seeding evaluation by RDI method include echo size, duration, and intensity. This study presents statistical analyses of the reflectivity pixels of the seeded clouds and the variation in *CAPPI* at the height of the seeding

layer. Since the seeding altitude is ~2 km and taking the vertical diffusion of hygroscopic particles into consideration, five levels of gridded *CAPPI* data are selected (from 1000 to 3000 m in 500m intervals). Quantities calculated at each level include the anomaly percentage of the gird number of effective echoes ($\sigma$), the maximum and mean echo intensity (*ref_max* and *ref_mean*), and the fractional contribution to the total reflectivities ($FCR_i$). *VIL* in the three domains are also calculated.

Although the RDI method could be used to evaluate the effect of cloud seeding by analyzing the variation in echo parameters

over a wide area during and after the seeding, it still has limitations. For example, the targeted cloud should ideally be stratiform with an embedded convective cloud, and the echo parameters of Domains A and B should better be homogeneous and linearly fluctuate to obtain stable systematic movement during the period of study so that the data have good consistency and small standard deviations. Additionally, the targeted cloud must be seeded as sufficiently as possible.

### 3.2 Results from the RDI algorithm

Raw data from the Zhoushan-based S-band Doppler Radar were obtained as input to the RDI algorithm to evaluate the efficacy of cloud seeding. Through analyzing the Shanghai, Hangzhou, and Taizhou's sounding data on 0400 UTC, we calculated the wind speed and direction at the altitude of seeding layer (Figure S1). By taking into consideration of the change of wind field



during and after cloud seeding, the direction in which the target radar echo moved and its speed were approximated. Table 2 defines the domains for evaluation purposes.

Figure 4 shows the time series of $\sigma$ and *ref_mean*. There is no significant variance in $\sigma$ ($\pm 1.59\%$), and the *ref_mean* is ~9.1 dBz ($\pm 0.3$ dBz) in Domain A during the intensive operation period (0300~0448 UTC), but $\sigma$ shows a gradual reduction in domain B (-15.02%) and C (-25.25%). Especially in Domain C, $\sigma$ value dramatically decreased after cloud seeding ended. Similarly, the *ref_mean* of Domain B and C fell to ~8 dBz at 0400 UTC, which was even lower than the background value (*ref_mean* of Domain A). The variation of echo intensity in Domain B and C is beyond that caused by natural variability (Domain A), which can be concluded that cloud seeding played a significant role in it. From another side, this comparison of $\sigma$ and *ref_mean* in the three domains suggests that the cloud seeding of an individual convective cell embedded in a widespread stratiform cloud contributes little to the natural variability of radar echoes at large scales. The dissipation of the seeded cloud, as evidenced by the variation in $\sigma$ and *ref_mean* in Domain B and C, is preliminarily considered the result of excessive seeding by hygroscopic particles.

Figure 5 shows the time series of the fractional contribution to the total reflectivities ($FCR_i$, where $i$ represents the $i$th reflectivity bin: $\leq 0$ dBz, 0–10 dBz, 10–20 dBz, 20–30 dBz, $\geq 30$ dBz) in Domain A, B, and C. The $FCR_i$ in these domains can be compared to assess differences in ranges. There was little variation in reflectivity across all bins in Domain A (Figure 5). About 80% of the grid points in this domain had reflectivity values between 10 and 30 dBz. However, $FCR_{(20dBz<i<30dBz)}$ and $FCR_{(i\geq30dBz)}$ decreased gradually over time in Domain B and C. In particular, after cloud seeding, $FCR_{(i\geq30dBz)}$ decreased to a minimum (~0%) in Domain C. The influence of cloud seeding on $FCR_{(10dBz<i<20dBz)}$ is not significant. The time series of $FCR_{(i<10dBz)}$ show different trends: an almost invariant trend in domain 1 (between 58.2% and 62.6%), a slight increasing trend in domain B (from 48% to 66%), and a stronger increasing trend in domain C (from 27% to 69%). These results suggest that cloud seeding effectively weakens the development of strong echoes (> 20 dBz). Meanwhile, accompanied by the seeding process, the appearance of weak echoes (< 10 dBz) increases.

The cloud layer in the lower troposphere moved from the northeast to the southwest. The cloud seeding started at 0336 UTC, and a corresponding maximum reflectivity of over ~35 dBz was seen (Figure 6a). From the vertical cross-section along the violet line in Figure 6a, there were strong echoes near the surface, and the echo top height was ~4.2 km (Figure 6c). As the seeding operation continued, *CR* and *VIL* decreased in varying degrees, and the targeted cloud gradually dissipated (see Figures S3 and S4). About 12 min after seeding ended (0418 UTC), *CR* decreased to a minimum (~10 dBz) and *VIL* was ~0.2 kg m$^{-3}$ (Figure 6e). The echo top height dropped to ~3.5 km, and the surface echoes also weakened.

### 3.3 Echo-cluster tracking and identification algorithm

For comparisons with the RDI results, an echo-cluster tracking and identification algorithm was launched to evaluate seeding efficacy. Details about the algorithm and uncertainties have been described elsewhere (Rosenfeld, 1987; Dixon and Wiener, 1993; Woodley and Rosenfeld, 2004). The echo volumes and mean CR from 0100 to 0600 UTC, i.e., the three-hour period





around the seeding time, were examined to identify convective cells in Domain A. Using a reflectivity threshold of $CR$ (~19 dBz), another four cells were identified (Figure 7).

Figure 8 shows the time series of echo volumes and mean $CR$s of the four unseeded cells. Both time series show the evolution and echo intensities of these cells. The mean $CR$s of the identified cells are roughly the same (the seeded cell is ~26.2 dBz, and the mean value of the other identified cells is ~25.2 dBz). However, the seeded cell appears to have the shortest life cycle (~1h 6min) compared with the other unseeded cells (mean value of the four cells is ~1h 46min).

### 3.4 Hourly variability of surface precipitation

The end result of cloud seeding often is associated with the variation of surface precipitation. If rainfall occurred, the surface echoes might weaken (red ovals in Figure 6) due to a natural process of cloud depletion. But from the echo-cluster tracking result between seeded and unseeded cells in Section 3.3, we can conclude that the seeded echo was weakened as the fastest speed and have the shortest life cycle. Figure 9 shows the rainfall distribution over cloud seeding region during 0200-0500 UTC, and the different values between them. The inhomogeneous feature of this precipitation event was also seen. The rain gauge data used in study consist of hourly precipitation with a space interval of ~10 km. Hourly precipitation during cloud seeding (0300-0400 UTC) with a maximum ~5.8 mm was stronger than that before (0200-0300 UTC) and after (0400-0500 UTC) cloud seeding. There were ~23 rain gauges with effective precipitation record (hourly precipitation ≥ 0.1mm) at 0400-0500 UTC, which was approximately one half of that at 0300-0400 UTC (~ 44 rain gauges). To better see the contrast of surface precipitation caused by cloud seeding, the interpolated 0.01° × 0.01° precipitation fields were constructed from site measurements to produce a time series of precipitation for each grid square. From the grid difference of precipitation between 0300-0400 UTC and 0400-0500 UTC, cloud seeding seems to have led a decrease in precipitation from seeded cloud relative to the surrounding clouds (Figure 9d).

### 3.5 Microphysical characteristics of the seeded cloud

Generally, Radar data can provide information about large hydrometeors such as raindrops. However, the hygroscopic flare used in this seeding experiment was mostly comprised of submicron and micro-hygroscopic particles (Table 1) which could directly affect the concentration and spectrum of cloud droplets. Calibrated airborne CDP measurements can provide such information about cloud droplets. Table 3 summarizes the microphysical characteristics of the targeted cloud during the various seeding periods (black box in Figure 2) and post-seeding sampling of the cloud layer (black oval in Figure 2). Stages I, II, and III represent the cloud seeding periods with an average flight altitude of 1875–1975 m, and stage IV represents the post-seeding sampling period when the aircraft flew through the seeded cloud again on its return to base. The mean flight altitude during stage IV was ~2200 m.

The cloud droplet number concentration ($N_c$), the effective diameter ($ED$), the liquid water content ($LWC$), and the cloud



droplet spectral dispersion ($\varepsilon$) increased in progression from stage I to stage III. The maximum $N_c$ increased from 216.7 cm$^{-3}$ to 322.4 cm$^{-3}$, and the mean $LWC$ increased from 0.4 g m$^{-3}$ to 0.8 g m$^{-3}$. Affected by hygroscopic particles, $N_c$ showed a bi-modal size distribution (peaks at 4–6 μm and 17–18 μm) during the seeding period (Figure 10). Large amounts of small particles (likely agent particles) were captured in the initial stage of cloud seeding (see the black arrow in the uppermost left-hand panel of Figure 10). Considering the potential hygroscopic growth of agent particles, the cloud number concentration of the first peak diminished and the second peak gradually increased. The spectrum also confirms that larger-mode (corresponding to second peak) particles were increased but showed slowing growth (17.9μm to 18.2μm) from $ED$ information in stage II and III. It can be concluded that some of the hygroscopic particles grow to cloud drops, even raindrops, through collision-coalescence process. But most of the agent particles were accumulated at 17-18μm by hygroscopic growth and hardly continues grow even bigger, which cloud be explained by the competition effect of water vapor. The $N_c$, $ED$ and $LWC$ were much lower and the drop size distribution was broadened of the seeded cloud during post-seeding sampling, likely because the seeding agents became progressively more dilute as the particles grow by condensation or dissipate by turbulent motion. More in-depth quantitative analyses are needed to examine this.

## 4. Conclusions

The goal of this study is to evaluate the potential effect of hygroscopic seeding on cloud and precipitation processes for the sake of suppressing rainfall during a major international event, the G20 Hangzhou Summit, around the southeastern coast of Zhejiang province in China. A marine stratocumulus cloud deck with a large horizontal extent was observed off the coast of eastern Zhejiang on 4 September 2016. Hygroscopic flares were dispersed into an appropriate region of the targeted cloud by an MA-60 research aircraft. Radiosonde soundings, real-time satellite images, radar data, and airborne CDP observations were all acquired to help identify cloud conditions suitable for cloud seeding. After seeding, the research aircraft flied into seeded cloud to measure cloud microphysical parameters.

Analysis of the differences in numerous cloud and rainfall parameters before and after seeding is a means for assessing the effect of the hygroscopic agents introduced into a convective cell embedded in a stratiform cloud. The marine stratocumulus clouds chosen in this study are under an enrichment condition of plentiful of water vapor, favorable for cloud development with rich CCN. By introducing hygroscopic agents into a small region of the targeted cloud and comparing its evolution with surrounding clouds in the same cloud regime, the role that other factors may have in modifying the cloud can be minimized so that the influence of cloud seeding can be singled out. A method for estimating the effectiveness of the seeding based on various parameters of radar echoes is presented. It is demonstrated that cloud seeding had altered the course of cloud development and their parameters, and suppressed precipitation.

An RDI algorithm based on radar grid data was proposed to analyze the cloud seeding effect. Echo tracking method including TREC technique and threshold strategy were performed for domain selection. Echo reflectivity parameters such as $ref\_mean$, $\sigma$, and $FCR_i$ were analyzed during and after the seeding process to examine any spatial differences. Results show that about 12





min after seeding, the composite reflectivity of the seeded cloud decreased to a minimum (< 10 dBz), and the *VIL* of the seeded cloud was ~0.2 kg m$^{-3}$. The echo top height dropped to ~3.5 km, and the surface echoes were also weakened. By contrast, there was no significant variation in the echo parameters of non-seeded clouds. The RDI results suggest that the hygroscopic seeding effectively weakened the development of strong echoes (> 20 dBz). After cloud seeding, the area of weak echoes (< 10 dBz) increased. The seeded echo had the shortest lifetime compared with the neighboring unseeded echoes identified by an echo-cluster tracking and identification algorithm. From airborne CDP measurement during cloud seeding, a small number of hygroscopic particles grow to cloud drops and raindrop through collision-coalescence process. However, most of the agent particles were accumulated at 17-18μm by hygroscopic growth and hardly continue bigger, presumably due to the competition effect of water vapor. It seems plausible that hygroscopic seeding creates competition mechanism and limits cloud development, thus suppresses precipitation.

It is admitted that this is just a case study with a certain degree of coincidence, from which we may hardly draw any solid conclusion that the change was totally due to seeding effect not natural variation. More observational evidences are certainly needed. They are, however, very costly and difficult to acquire especially in densely populated regions where access of aerospace is usually extremely difficult to gain in order to fly into the right clouds at the right time. In this regard, the case as studied here is an invaluable sample that is worth exploring.

*Data availability.* All observations data used in this study are available. Readers can access the data directly or by contacting Fei Wang via feiwang@cma.gov.cn.

*Author contribution.* F. Wang and Y. Zhou designed the aircraft campaign; F. Wang, G. Wang analyzed the radar data; F. Wang, Q. Jiang, J. Duan and S. Jia analyzed satellite, rain gauge and airborne data; F. Wang and Z. Li wrote the paper.

*Acknowledgements.* This study was supported by National Key Research and Development Program of China (2016YFA0601701 and 2017YFC1501702) and the National Science Foundation of China (91544217). We are very grateful to the reviewers for their constructive comments and thoughtful suggestions. We also thank all of the experimental research team, especially the flight crew of CMA's MA-60 airplane. The observational data used in this paper for radar, precipitation, radiosonde can be accessed from the website http://data.cma.cn/. The airborne data of the field experiment also available. Readers can access the data by contacting feiwang@cma.gov.cn.





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



**Table 1.** Technical parameters of the ZY-1NY cloud seeding flare. (The information of ZY-1NY flare was mainly extracted from http://www.zthj.com/product/info/11)

| Combustion product of Hygroscopic Flare | Agent dosage | Combustion time | Seeding rate | | | |
|---|---|---|---|---|---|---|
| | | | All | D >0.5μm | D>0.9μm | D>2μm |
| KCl, CaCl$_2$ | 1.2 kg | 15–18 min | $4.4\times10^{13}$/s | $4.0\times10^{11}$/s | $6.0\times10^{10}$/s | $2.0\times10^{10}$/s |

**Table 2.** Definition of domains for the RDI method.

| Domain | Dimension | Item | Feature |
|---|---|---|---|
| A | 4°×4° | Effective detection scope of the radar | Immovable |
| B | 0.5°×0.5° | Possible region affected by cloud seeding | Immovable |
| C | 0.1°×0.1° | Target echo corresponding to the seeding cloud | Movable |

**Table 3.** Statistics (mean and maximum values) of airborne CDP-measured microphysical parameters (cloud droplet number concentration $N_c$, effective diameter $ED$, liquid water content $LWC$, and cloud droplet spectral dispersion $\varepsilon$) on 4 September 2016 for seeding and post-seeding legs at approximately the cloud seeding height.

| Stage | Status | Altitude | $N_c$ (cm$^{-3}$) | | $ED$ (μm) | | $LWC$ (g kg$^{-1}$) | | $\varepsilon$ | |
|---|---|---|---|---|---|---|---|---|---|---|
| | | | Mean | Max | Mean | Max | Mean | Max | Mean | Max |
| I | Seeded | 1975.4 | 86.2±54.1 | 216.7 | 15.0±6.6 | 26.2 | 0.4±0.3 | 1.9 | 0.2±0.1 | 0.4 |
| II | Seeded | 1875.4 | 76.5±53.2 | 240.1 | 17.9±2.4 | 25.0 | 0.6±0.4 | 2.2 | 0.3±0.1 | 0.5 |
| III | Seeded | 1874.6 | 119.6±103.5 | 322.4 | 18.2±4.1 | 23.4 | 0.8±0.7 | 2.8 | 0.3±0.1 | 0.5 |
| IV | Unseeded | 2102.9 | 38.1±12.5 | 57.1 | 14.1±1.7 | 26.4 | 0.2±0.1 | 0.4 | 0.4±0.1 | 0.6 |





**Figure 1.** Cloud image of Himawari-8 at visible channel (a), cloud top height (b), and cloud optical depth (c) at 0300 UTC. The rectangles in b and c indicate the experimental region.



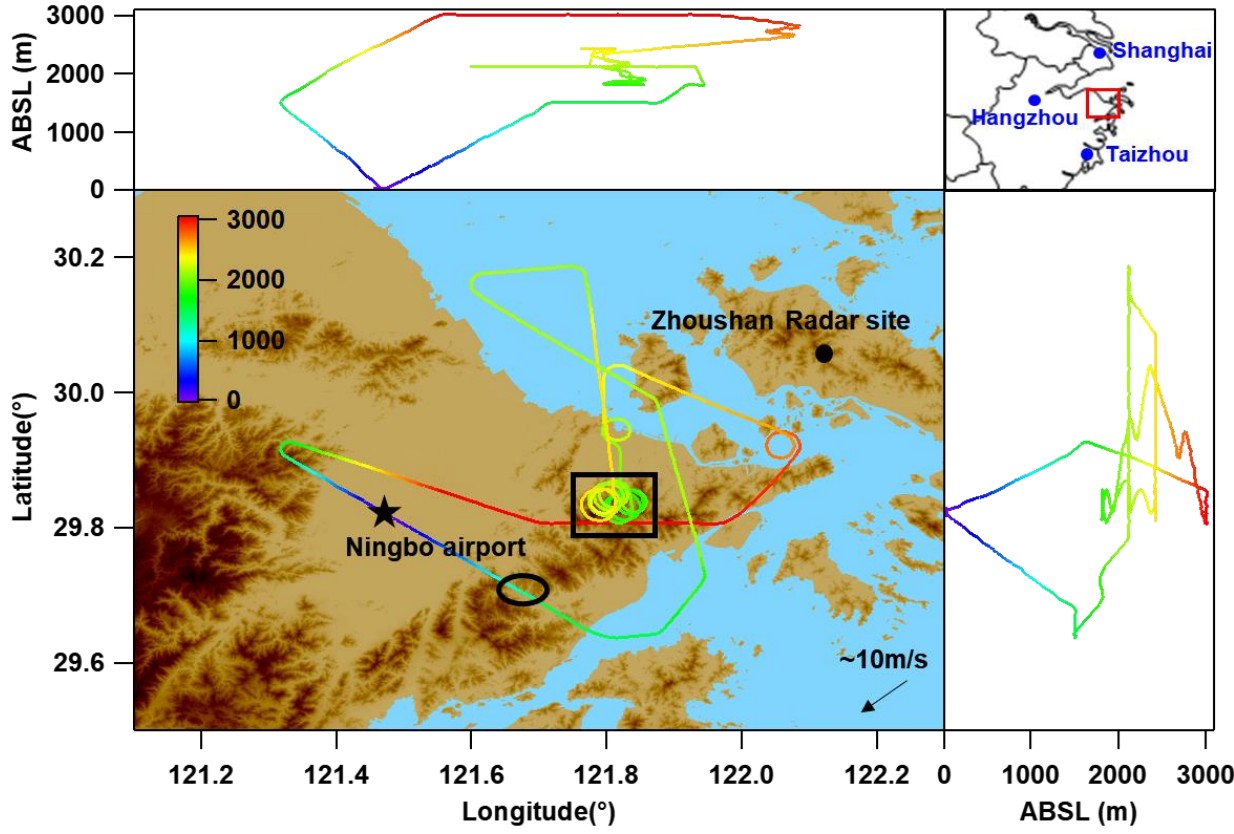

**Figure 2.** Geographical location of the experimental region (top right: the red box denotes the flight region, the blue spots denote the Shanghai, Hangzhou, and Taizhou radiosonde station) and the flight track of the research flight. The black box shows the area of cloud seeding, and the black oval shows the area where post-seeding observations of the seeded clouds were made.





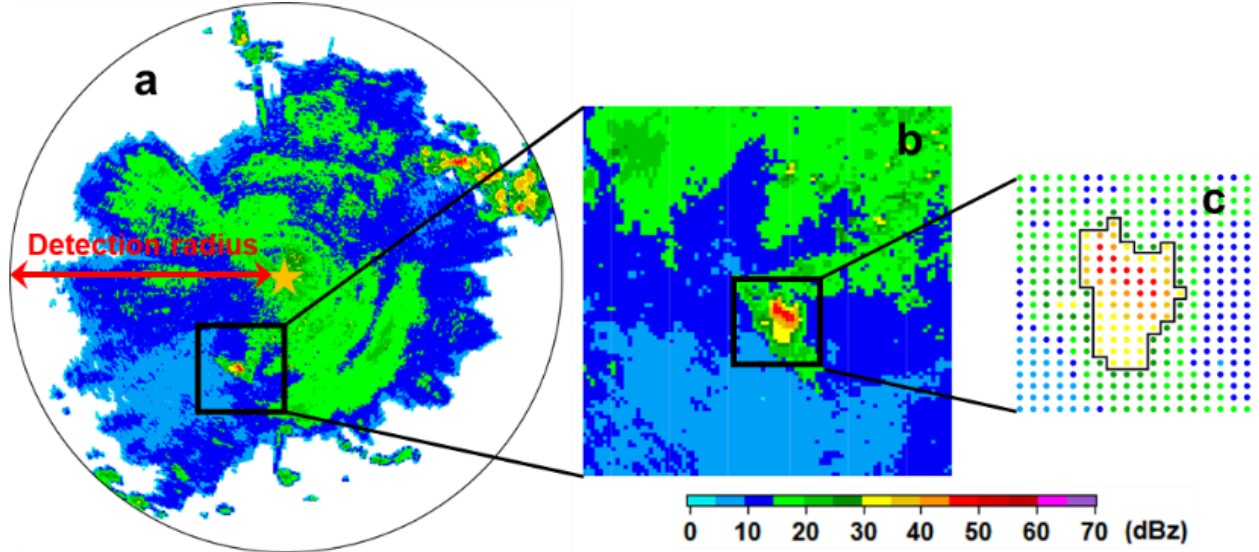

**Figure 3.** Schematic diagrams showing how the RDI method selects a domain. Take single radar as example. (a) The background radar echo field is defined as Domain A. The orange star shows the location of the radar site. The effective detection scope (corresponding to Domain A) is ~5°×5°. (b) The black rectangle outlines Domain B, representing a ~0.5°×0.5° or ~1°×1° polluted region that was affected by seeding agents based on the motion of the seeded cloud or seeding plumes. (c) Domain C is a movable region using the threshold strategy and the TREC technique. The background field (Domain A) and the polluted region (Domain B) could be adjusted according to the scope of the seeding plume such as the regional mosaic reflectivity field retrieved by multiple radars.





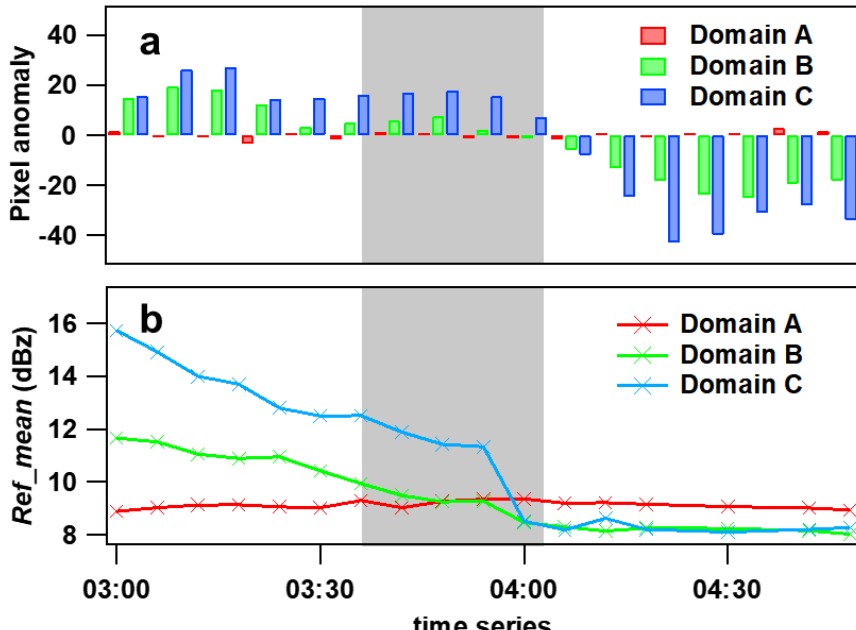

**Figure 4.** Time series of (a) the anomaly percentage of grid number of effective echoes and (b) mean echo intensity (*ref_mean*) in Domain A, B, and C. The five *CAPPI* levels are considered. Gray shaded areas show when cloud seeding was done.





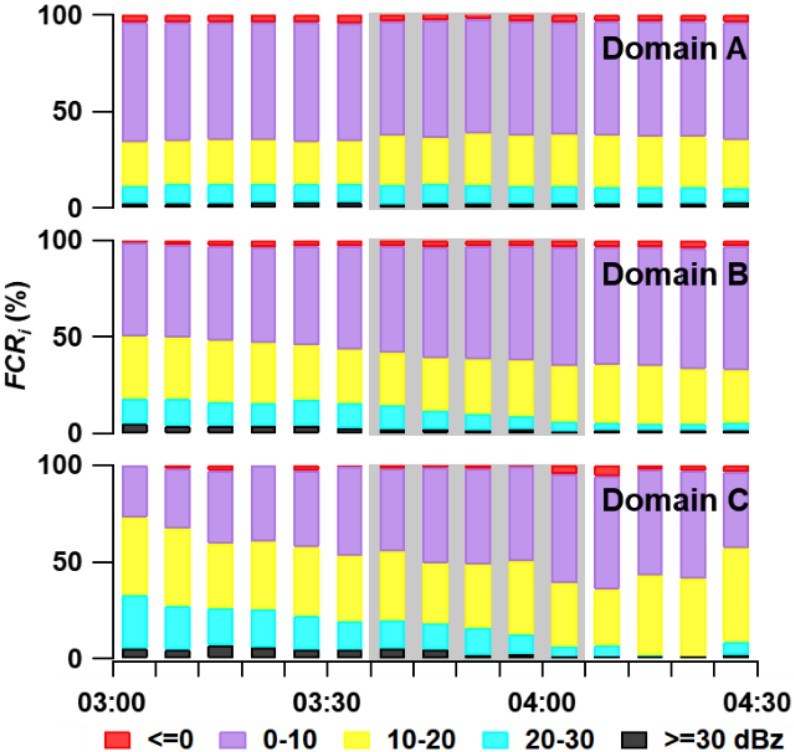

**Figure 5.** Time series of the fractional contribution to the total reflectivities ($FCR_i$, where $i$ represents the $i$th reflectivity bin: $\leq 0$ dBz, 0–10 dBz, 10–20 dBz, 20–30 dBz, $\geq 30$ dBz) in Domain A, B, and C. The five *CAPPI* levels are considered. Gray shaded areas show when cloud seeding was done.




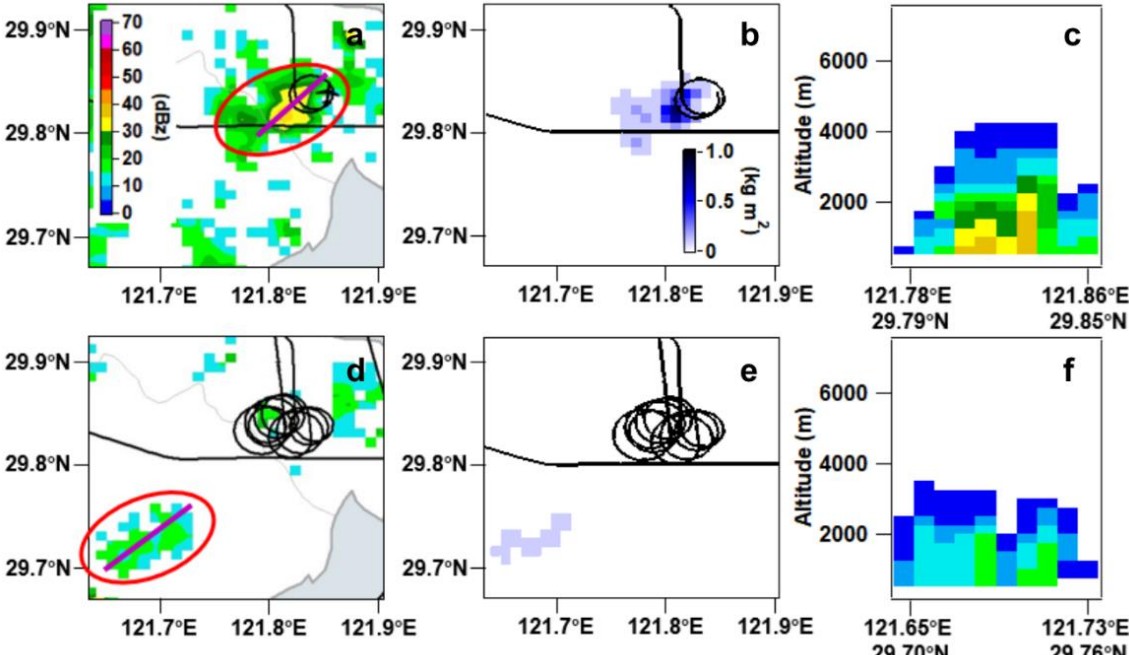

**Figure 6.** (a, d) Composite reflectivity from the five levels (1000–3000 m in 500-m intervals), (b, e) vertically integrated liquid water content, and (c, f) vertical cross-section [along the violet lines in (a, d)] of the seeding echo. The top panels are for the start of cloud seeding (0336 UTC), and the bottom panels are for 12 min after the end of seeding (0418 UTC). The red ovals in (a, d) outline the seeding cloud, and the black lines show the flight tracks. To clearly show the seeding cloud in (a, d), weak echoes (< 10 dBz) are rejected.



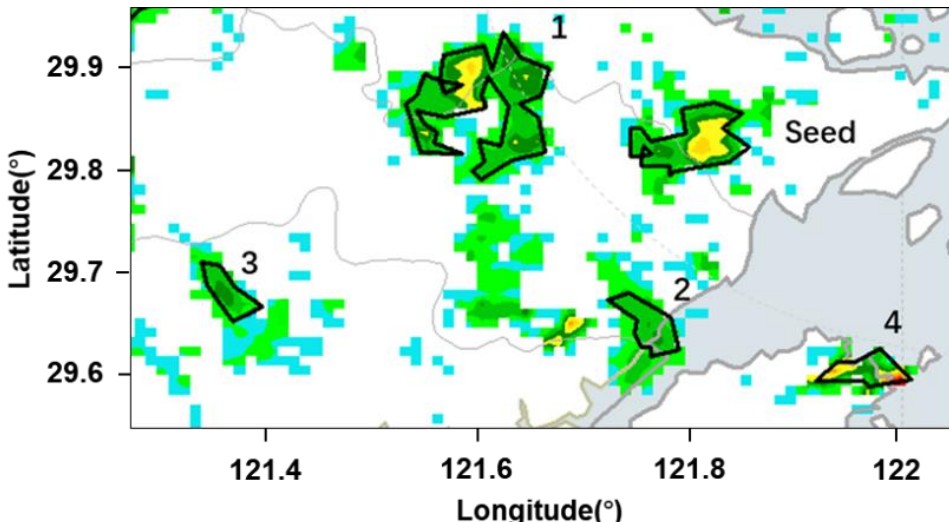

**Figure 7.** Composite reflectivity of the seeded cell and other cells identified by the cloud cluster algorithm at 0336 UTC (cell numbers 1–4). The reflectivity threshold is ~19 dBz. To clearly show identified cells, weak echoes (< 10 dBz) are rejected.





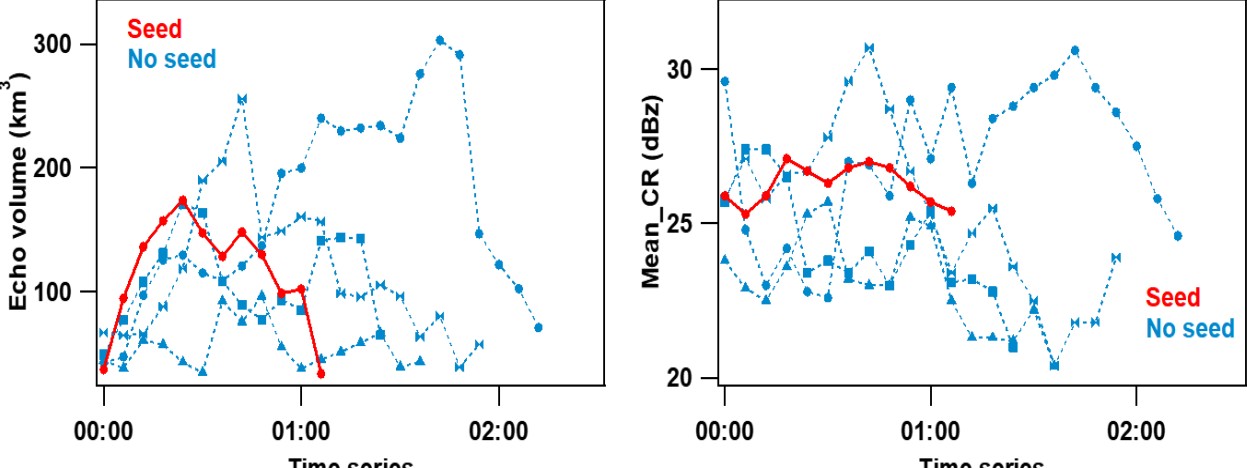

**Figure 8.** Echo volume (left panel) and mean composite reflectivity (right panel) of identified cells (including seeded and

unseeded cells). All sampling cells are normalized to the same initial time on the x-axis.



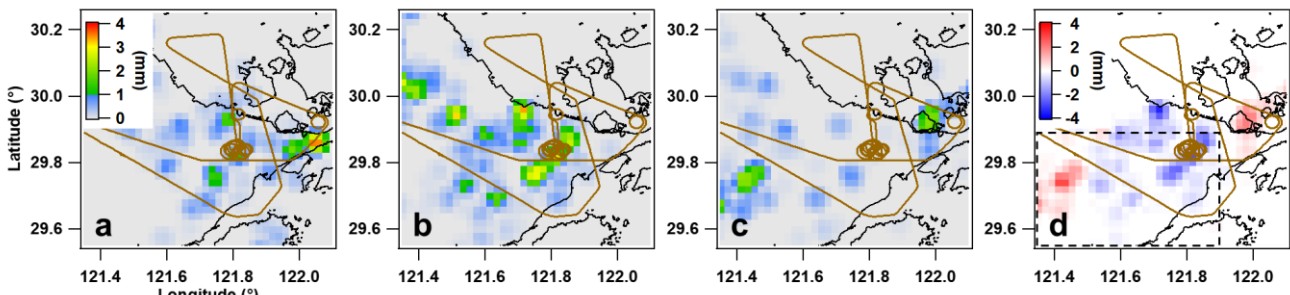

**Figure 9.** Hourly surface precipitation at 0200-0300 UTC (a), 0300-0400 UTC (b), and 0400-0500 UTC (c), which approximately consider as before, during, and after cloud seeding. The right graph (d) indicates the grid difference of precipitation between 0300-0400 UTC and 0400-0500 UTC.

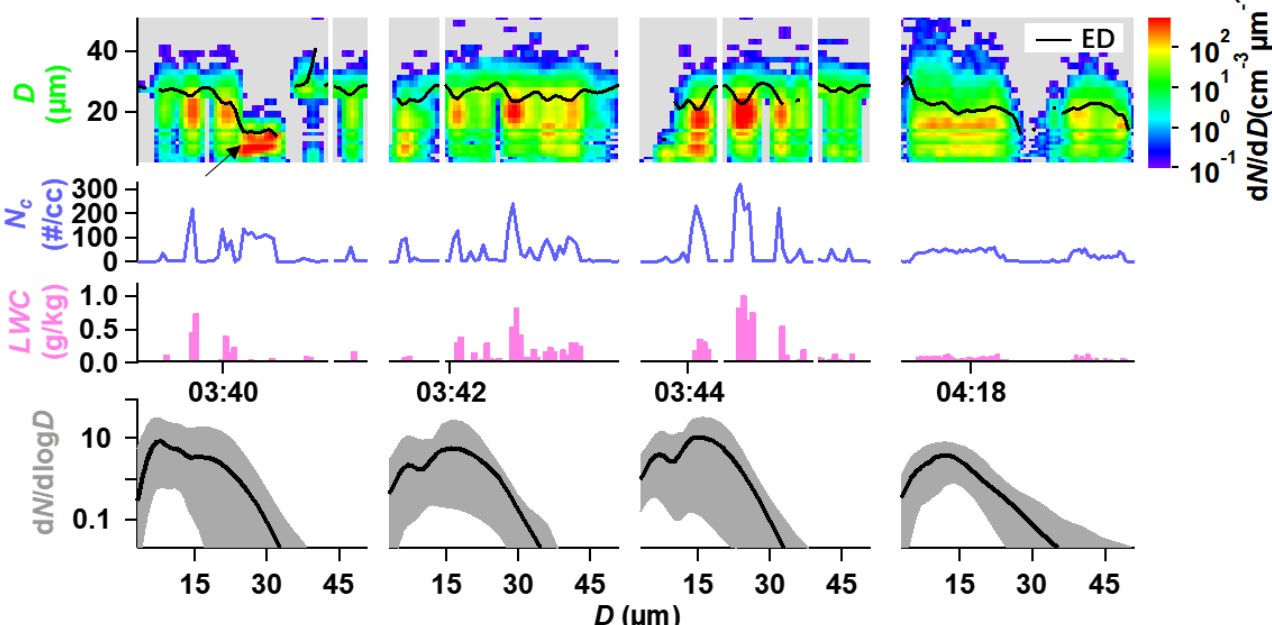

**Figure 10.** Top three rows: The time series of (row 1) cloud number size distribution ($dN/dD$; color shaded) and effective diameter ($ED$; black solid line), (row 2) cloud droplet number concentration ($N_c$), and (row 3) liquid water content ($LWC$). Bottom row: Mean spectrum during the flight though the seeded cloud measured by an aircraft-mounted CDP. Black lines represent the means and the gray shaded areas represent the 10th and 90th percentiles of the data. The first three columns from the left represent the seeding period (corresponding to the black box in Figure 2) and the rightmost column represents the period when the aircraft flew through the seeded cloud again on its return to base (corresponding to the black oval in Figure 2).