# Peer review of "Evaluation of hygroscopic cloud seeding in liquid-water clouds: a feasibility study"

_Atmospheric Chemistry and Physics, 2019_

## Referee Comment (RC1) · Anonymous Referee #1 · 15 Jun 2019

This study attempts to estimate the hygrpscopic seeding in liquid water clouds by combining the observations from radar, surface precipitation and airborne Cloud Droplet Probe (CDP) measurement. Their findings in this study confirm that cloud seeding had altered the course of cloud development and their parameters, and suppressed precipitation. Although this study only focuses on the case study, the related results may help for understanding about the course of hygroscopic seeding and provide a feasibility method for cloud seeding assessment. Generally speaking, the paper is interesting, and tables and graphics are well constructed. The objectives of the study are clearly mentioned in the introduction and the results associated with their significances are stated in the conclusion. As a result, I am recommending the paper be accepted with minor revisions if the authors response properly my comments. The some main com-

ments and suggestion I have are listed below in the specific comments to the authors.

Specific Comments:

1. Page 1, Line 24: CDP—>Cloud Droplet Probe (CDP).

2. Figure 1(b), the unit of CTH—>km.

3. Page 3, Line27-28: This study only focused on the warm liquid water cloud, right? Are the supercooled water clouds excluded in the analysis?

4. For the Figure 2, a detailed information is needed in the figure caption. Such as, what's the mean of the top left and bottom right subplots in the figure 2.

5. Page 6, Line 19: domain 1——>domain A.

6. Page 7, Line 1: the reflectivity threshold of CR is 19dBz, please give the reference.

7. Page 7, Line 5: I am a little bit confused by your results in the Figure 8. In my feeling, the cloud seeding may alter cloud properties and suppress precipitation, it means that clouds should have more longer lift time. But, the seeded cell appears to have the shortest life cycle in your study, why?

8. Please add the color bar in the Figure 6c, 6f and Figure 7.

---

## Referee Comment (RC2) · Anonymous Referee #2 · 15 Sep 2019

General comments: Assessment remains elusive of the hygroscopic cloud seeding, given the buffering effect induced by the co-variability of meteorology, aerosol particles (agents) and clouds. The authors comprehensively evaluated the effect of hygroscopic cloud seeding used a combination of aircraft measurements, ground-based radar and gauge observations, along with radiosonde and geostationary data. The topic is of interest and the data obtained here are valuable to the weather modification community. However, the paper is not well organized and should be substantially revised before it can be acceptable for publication. Also, the writing should be polished to improve the readership given a lot of grammar errors. Below listed are the major comments and minor comments returned to the authors for consideration.

Major comments: 1. Major findings are derived from ground-based radar data and

aircraft-based CDP measurements, which is valuable to the scientific community. Given that Himawari is new-generation geostationary satellite which provides cloud observations at 10-min intervals, Figure 1b-c is a good case in point. I am wondering whether the authors can show any images of post-seeding, in order to provide circumstantial evidence for the argument of "The echo top height dropped to ∼3.5 km". Are there any changes in the cloud top height as observed from Himawari? I think this addition and related discussion will make this paper more convincing. 2. The organization of this manuscript needs to be substantially changed. It is customary to put "methods" together with "data" rather than with "results". Therefore, sections 3.1 and 3.3 are advised to be moved to section 2. On top of it, the title of section 3 can be revised to "Results and discussion". As such, more discussion is required in analyzing the observational results to enhance its readership. 3. Sections 3.4, and 3.5: In the domain with cloud seeding, it seems to me that the precipitation peaked at 0300-0400 UTC from the perspective of life cycle of the stratocumulus clouds analyzed here. This will undoubtedly result in the expected results shown here. It is therefore supposed to add some discussion in this regard.

Minor comments: 1. Page 1 Line 26: Several grammar errors in "This probably because the hygroscopic growth by agent particles and collision-coalescence by small cloud.." 2. Page 1 Line 26: There is a typo here. "is" is missing in "which probably" 3. Page 1 Line 1: something is missing after "ground-based" 4. Page 2 Line 30: the author may consider to add "ending up with delayed onset of precipitation (Rosenfeld et al., 2014; Guo et al., 2016; Lee et al., 2016)" following "precipitation (Rosenfeld et al., 2008)." 5. Page 3 Lines 7-8: It is advised to mention several most recent assessment studies conducted in China used ground-based radar and aircraft measurements, including Wang et al.. 2019. J. Meteor. Res., doi: 10.1007/s13351-019-8122-1. 6. Page 3 Lines 14-16: The three radiosonde sites are part of radiosonde observational network operated by CMA, which is supposed to be mentioned here. Also, the uncertainties of humidity and temperature are needed to be discussed, given they have been used to derive the vertical structures of clouds. 7. Page 5 Line 11: "which pending"-> "depending" 8. Page

5 Line 31: 0600 UTC ? 9. Page 7 Line 10: "as"->"at" 10. Page 8 Lines 19-20: "Radiosonde soundings, real-time satellite imagesc, and airborne CDP observations were all acquired to help identify cloud conditions suitable for cloud seeding." is misleading, especially for the purpose of radiosonde. I noticed in the supplementary materials, the authors used 0600 UTC, which was 2-3 hrs after cloud seeding. 11. Page 9 Line 12: "not"-> "rather than" or "instead" 12. The authors may consider to revise the x-axis title from "Time series" in Figures 4 and 8 to "Hours (UTC)" 13. The caption in Figure 9: "consider" is advised to be changed to "corresponds to" ; "right graph" is advised to be changed to "rightmost panel"

References:

Rosenfeld D, et al., 2014. Climate effects of aerosol-cloud interactions. Science. 343(6169):379-80. Guo, J. et al. 2016, Delaying precipitation and lightning by air pollution over the Pearl River Delta. Part I: Observational analyses, J. Geophys. Res. Atmos., 121, 6472–6488, doi:10.1002/2015JD023257. Lee, S.-S., et al., 2016. Delaying precipitation by air pollution over Pearl River Delta. Part 2: model simulations, J. Geophy. Res. Atmos., 121, 11,739–11,760, doi: 10.1002/2015JD024362 Wang, W., et al. 2019. Extra-area effect in operational cloud seeding during winter in Jiangxi province, East China. J. Meteor. Res., 33(3), doi: 10.1007/s13351-019-8122-1.

———————————————

---

## Author Comment (AC1) · 31 Oct 2019

Please see the attached PDF files for our response and the revised manuscript.

Please also note the supplement to this comment:
https://www.atmos-chem-phys-discuss.net/acp-2019-408/acp-2019-408-AC1-supplement.zip

---

## Author Response (AR1)

**Anonymous Referee #1:**

We sincerely appreciate for your professional review work on our paper. The comments and suggestions you gave are very helpful for us to improve our study. Below are our point-by-point responses (in black) to all your comments (in green) and the corresponding changes in the revised manuscript are highlighted in blue.

**Specific comments:**

1. *Page 1, Line 24: CDP -> Cloud Droplet Probe (CDP).*

It has been corrected in the revised manuscript.

2. *Figure 1(b), the unit of CTH -> km.*

It has been corrected in the revised manuscript.

3. *Page 3, Line27-28: This study only focused on the warm liquid water cloud, right? Are the supercooled water clouds excluded in the analysis?*

Yes, we only discuss water cloud in this study. From radiosonde and airborne data analysis, we conclude that the low-level cloud, which aim to cloud seeding, was mainly composed of liquid phase.

4. *For the Figure 2, a detailed information is needed in the figure caption. Such as, what's the mean of the top left and bottom right subplots in the figure 2.*

It has been added in the revised manuscript.

5. *Page 6, Line 19: domain 1 -> domain A.*

It has been corrected in the revised manuscript.

6. *Page 7, Line 1: the reflectivity threshold of CR is 19dBz, please give the reference.*

There is no exact reference about the selection of reflectivity threshold. Our goal was to select a lower limit of reflectivity, which could objectively and completely characterize the life history of seeded and non-seeded clouds. We tried different values in the range of 15~25 dBz, and 19dBz was the final choice.

7. *Page 7, Line 5: I am a little bit confused by your results in the Figure 8. In my feeling, the cloud seeding may alter cloud properties and suppress precipitation, it means that clouds should have more longer lift time. But the seeded cell appears to have the shortest life cycle in your study, why?*

First of all, the decaying of seeded cloud cell was observed by a rain radar which measures rain drops only. The shortest life cycle of radar echo does not mean that the life time of this cloud is also the shortest.

Secondly, we totally agree with you that cloud seeding may alter cloud properties and suppress precipitation, which lead a longer life time. In this study, a large number of cloud droplets were

observed accumulating to 17-18μm by hygroscopic growth and few of them continue grow bigger (figure 10), which confirmed your viewpoint from another aspect. However, this process only lasts for a short while, probable due to the initiation of precipitation that dissipates cloud. In our opinion, compared with the natural process of cloud depletion, the increase of cloud life time by cloud seeding was much weaker.

*8. Please add the color bar in the Figure 6c, 6f and Figure 7.*

It has been added in the revised manuscript.

**Anonymous Referee #2:**

We sincerely appreciate for your careful review of our paper. The comments and suggestions you gave are very helpful for us to improve our study. Below are our point-by-point responses (in black) to all your comments (in green) and the corresponding changes in the revised manuscript are highlighted in blue.

**Major comments:**

9. *Major findings are derived from ground-based radar data and aircraft-based CDP measurements, which is valuable to the scientific community. Given that Himawari is new-generation geostationary satellite which provides cloud observations at 10-min intervals, Figure 1b-c is a good case in point. I am wondering whether the authors can show any images of post-seeding, in order to provide circumstantial evidence for the argument of "The echo top height dropped to ~3.5 km". Are there any changes in the cloud top height as observed from Himawari? I think this addition and related discussion will make this paper more convincing.*

According to your suggestion, we analyzed cloud top height (CTH) before, during and after cloud seeding from satellite image (figure 1), trying to find any evidence of CTH change by cloud seeding. Unfortunately, due to the dual-layer of cloud structure (see figure1 in the manuscript and S1), the seeded cloud was a small convective cell which was totally shaded by the upper-layer cloud. It was displayed as few pixels in the satellite image, from which one can hardly give any helpful information.

[Figure]

Figure 1 Cloud top height (CTH) of Himawari-8 at 0300 (a), 0400 (b), and 0500 (c), which cloud represent CTH before seeding, seeding, and post-seeding approximately.

10. *The organization of this manuscript needs to be substantially changed. It is customary to put "methods" together with "data" rather than with "results". Therefore, sections 3.1 and 3.3 are advised to be moved to section 2. On top of it, the title of section 3 can be revised to "Results and discussion". As such, more discussion is required in analyzing the observational results to enhance its readership.*

Per the suggestions (very good!), the title of section 2 and section 3 were revised to "Data and analysis method" and "Results and discussion". Correspondingly, we added "2.1 Experimental and data description", "2.2 Radar-domain-index (RDI) algorithm", "2.3 Echo-cluster tracking and identification algorithm" as sub-section of section 2. More specific discussions on "Evaluation by echo-cluster tracking and identification algorithm" and "Hourly variability of surface precipitation" were added. The corresponding changes in the revised manuscript have

been highlighted in blue.

*11. Sections 3.4, and 3.5: In the domain with cloud seeding, it seems to me that the precipitation peaked at 0300-0400 UTC from the perspective of life cycle of the stratocumulus clouds analyzed here. This will undoubtedly result in the expected results shown here. It is therefore supposed to add some discussion in this regard.*

We agree and the following discussion was added to section 3.3 in the revised manuscript.

"On the other hand, it seems that the precipitation peaked at 0300-0400 UTC from the analysis of cloud life cycle using radar echo in section 3.3. The decrease of surface precipitation is probably due to a natural process of cloud depletion. However, according to the seeding time, extent and dosage in this experiment, the hygroscopic seeding could just change the cloud number concentration and size distribution in a very limited scope. Besides, cloud seeding is a chain of physical process, similar to cloud-precipitation process in nature, and it influences surface precipitation through a complex mechanism. From our comparative analysis, its development was restrained and its life cycle was shortened, which was also demonstrated by micro-physical analysis from our airborne data. Actually, surface precipitation was observed no longer increasing in the domain following cloud seeding. We thus consider it a necessary but insufficient condition for the evaluation."

**Minor comments:**

*1. Page 1 Line 26: Several grammar errors in "This probably because the hygroscopic growth by agent particles and collision-coalescence by small cloud…"*

We have revised this as "This is probably caused by hygroscopic growth of agent particles and collision-coalescence of small cloud droplets."

*2. Page 1 Line 26: There is a typo here. "is" is missing in "which probably"*

It has been corrected in the revised manuscript.

*3. Page 2 Line 1: something is missing after "ground-based"*

We have revised it as "ground-based generator".

*4. Page 2 Line 30: the author may consider to add "ending up with delayed onset of precipitation (Rosenfeld et al., 2014; Guo et al., 2016; Lee et al., 2016)" following "precipitation (Rosenfeld et al., 2008)."*

It has been added in the revised manuscript.

*5. Page 3 Lines 7-8: It is advised to mention several most recent assessment studies conducted in China used ground-based radar and aircraft measurements, including Wang et al. 2019. J. Meteor. Res., doi: 10.1007/s13351-019-8122-1.*

It has been added in the revised manuscript.

*6. Page 4 Lines 14-16: The three radiosonde sites are part of radiosonde observational network operated by CMA, which is supposed to be mentioned here. Also, the uncertainties of humidity and temperature are needed to be discussed, given they have*

*been used to derive the vertical structures of clouds.*

They have been added in the revised manuscript.

There are two meanings of the uncertainties, the instrument itself and the algorithm of cloud identification. I think both of them were important in discussing cloud vertical structure. So, we add the following description in Section 1 in the supplement material.

In this study, we use RH profiles derived from sounding data, to analyze the vertical structure of clouds. First of all, the accuracy of RH from radiosonde observation is particularly important. The L-band sounding system, measured once per second, was widely used in operational radiosonde stations in China since 2002 (Zhang et al., 2016; Guo et al., 2016). The GTS1 digital electronic sensor, one of the key components of the L-band sounding system, provides fine-resolution profiles of temperature, pressure, RH, wind speed and direction at least twice a day to monitor the vertical profiles of atmospheric thermodynamic condition. Comparisons between GTS1 and Vaisala RS80 from previous studies indicated adequate agreements in the troposphere, but a lot larger biases in the upper atmosphere (Bian et al., 2011). Compared with Vaisala RS92, GTS1 sensor was found to yield a systematic dry bias in the order of 10% below 500 hPa. The GTS1 sensor showed a delayed response or a lag effect after the humidity changes rapidly like going through a cloud layer (Li et al., 2009).

Accurate identification of clouds by using sounding data is also import to this study. The method we choose may seriously affect the result. Three widely used algorithms have been employed to determine the locations of cloud layers and their boundaries from radiosonde observations, including:

a.    Dewpoint temperature depressions below certain threshold (Poore et al., 1995);

b.    Cloud detection method based on T (z) and RH (z), which are the second-order derivatives of temperature and RH with respect to height, respectively (Chernykh and Eskridge, 1996).

c.    RH thresholding method (Wang and Rossow, 1995), i.e. $RH_{max}>87\%$, $RH_{min}>84\%$, a RH jump at cloud base and cloud top; (here after WR95)

The main uncertainty of WR95 method is that it tends to misclassify moist, cloud-free layers as clouds. To avoid this, an improved algorithm was proposed by using W-band cloud radar, ceilometer and satellite observation (Wang et al., 1999; Zhang et al., 2010). In this study we use the WR95 method to determine cloud vertical structure, with the RH threshold being set to >84% according to the findings from previous inter-comparison studies (Zhou and Ou, 2010).

**7.   *Page 5 Line 11: "which pending"-> "depending"***

It has been corrected in the revised manuscript.

**8.   *Page 5 Line 31: 0600 UTC?***

Yes, the radiosonde was launched at 0600 UTC, it has been corrected in the revised manuscript.

**9.   *Page 7 Line 10: "as"->"at"***

It has been corrected in the revised manuscript.

*10. Page 8 Lines 19-20: "Radiosonde soundings, real-time satellite images, and airborne CDP observations were all acquired to help identify cloud conditions suitable for cloud seeding." is misleading, especially for the purpose of radiosonde. I noticed in the supplementary materials, the authors used 0600 UTC, which was-3 hrs after cloud seeding.*

As the reviewer noted, sounding data are helpful in identifying cloud vertical structure after the experiment. Our original statement was indeed misleading that was thus deleted in the revised manuscript.

*11. Page 9 Line 12: "not"-> "rather than" or "instead"*

has been corrected in the revised manuscript.

*12. The authors may consider to revise the x-axis title from "Time series" in Figures 4 and 8 to "Hours (UTC)"*

It has been revised in the revised manuscript.

*13. The caption in Figure 9: "consider" is advised to be changed to "corresponds to"; "right graph" is advised to be changed to "rightmost panel"*

It has been corrected in the revised manuscript.